# Structural characterization of the ICOS/ICOS-L immune complex reveals high molecular mimicry by therapeutic antibodies

Edurne Rujas[1,2], Hong Cui[1], Taylor Sicard[1,3], Anthony Semesi[1] & Jean-Philippe Julien [1,3,4 ✉]

The inducible co-stimulator (ICOS) is a member of the CD28/B7 superfamily, and delivers a positive co-stimulatory signal to activated T cells upon binding to its ligand (ICOS-L). Dysregulation of this pathway has been implicated in autoimmune diseases and cancer, and is currently under clinical investigation as an immune checkpoint blockade. Here, we describe the molecular interactions of the ICOS/ICOS-L immune complex at 3.3 Å resolution. A central FDPPPF motif and residues within the CC' loop of ICOS are responsible for the specificity of the interaction with ICOS-L, with a distinct receptor binding orientation in comparison to other family members. Furthermore, our structure and binding data reveal that the ICOS N110 N-linked glycan participates in ICOS-L binding. In addition, we report crystal structures of ICOS and ICOS-L in complex with monoclonal antibodies under clinical evaluation in immunotherapy. Strikingly, antibody paratopes closely mimic receptor-ligand binding core interactions, in addition to contacting peripheral residues to confer high binding affinities. Our results uncover key molecular interactions of an immune complex central to human adaptive immunity and have direct implications for the ongoing development of therapeutic interventions targeting immune checkpoint receptors.

[1] Program in Molecular Medicine, The Hospital for Sick Children Research Institute, Toronto, ON M5G 0A4, Canada. [2] Biofisika Institute (CSIC, UPV/EHU) and Department of Biochemistry and Molecular Biology, University of the Basque Country (UPV/EHU), P.O. Box 644, 48080 Bilbao, Spain. [3] Department of Biochemistry, University of Toronto, Toronto, ON M5S 1A8, Canada. [4] Department of Immunology, University of Toronto, Toronto, ON M5S 1A8, Canada. ✉email: jean-philippe.julien@sickkids.ca

Pathogenic infections trigger a series of highly regulated events orchestrated by the immune system to limit host damage and provide long-term protection. Upon encounter of foreign antigens, antigen-presenting cells (APCs) display specific peptide major histocompatibility complexes (MHC) on their surface that can be recognized by the T-cell receptors (TCR) of naïve antigen-specific T cells. However, this first signal is not sufficient for successful activation of T cells[1], and a second synergistic signal provided by interactions of co-stimulatory receptors on T cells is required with their cognate ligands in APCs[2].

A key molecule that provides this secondary signal and hence dictates T cell fate is the inducible T-cell co-stimulator (ICOS)[3] and its unique ligand (ICOS-L) on APCs[4,5]. ICOS (CD278) is a type I transmembrane glycoprotein that belongs to the CD28 family of co-stimulatory immunoreceptors. It is present on the T cell surface as a disulfide bond-linked homodimer[3] and it is rapidly upregulated upon TCR cross-linking and/or CD28 co-stimulation[3,5,6]. On the other hand, ICOS-L (CD275) belongs to the B7 family and is expressed on the surface of APCs[5,7,8] and non-hematopoietic cells under inflammatory conditions[9–12]. Binding of ICOS-L to ICOS triggers distinct intracellular signaling cascades through the conserved motifs YMFM[13], IProx[14], and KKKY[15] in the ICOS cytoplasmic tail. These signaling pathways deliver the co-stimulatory signals that promote T cell activation and differentiation.

Multiple studies support a multifaceted function for ICOS/ICOS-L and hence a complex role in dictating the course of adaptive immunity. Individuals with null mutations in the ICOS gene[16] and ICOS-deficient animal models[17–19] exhibit a profound defect in humoral responses due to the lack of T follicular helper (Tfh) cells, a specialized CD4[+] T cell subset essential for germinal center (GC) formation[20]. In addition to regulating thymus-dependent (TD) Ab responses, ICOS also affects Th1, Th2, and Th17 immunity[21,22] and the homeostasis of regulatory T cells (Treg)[23]. The expression of ICOS has also been reported in innate lymphoid cells (ILCs), which expands the role of this receptor to the innate arm of the immune system[24]. This multiplicity of roles underpins the relevance of the ICOS/ICOS-L signaling pathway and in turn, the tremendous potential of manipulating this co-stimulatory signal in the development of cancer immunotherapies and in the treatment of autoimmune diseases.

Various studies have demonstrated that anti-tumor T cell responses in mice can be significantly boosted by combining cytotoxic T lymphocyte-associated antigen 4 (CTLA-4) blockade with ICOS engagement[25,26]. In addition, emerging evidence suggests that ICOS blockade holds promise for the treatment of inflammatory diseases such as allergic asthma[24]. Anti-ICOS monoclonal antibody (mAb) therapy that blocks ICOS signaling has proven beneficial in the transplant field by inducing tolerance following cardiac allograft in rats[27–29]. As a consequence, the number of antibodies targeting the ICOS/ICOS-L immune complex entering clinical trials is rapidly increasing. In particular, the humanized monoclonal anti ICOS-L antibody prezalumab has recently shown efficacy in the treatment of patients with systemic lupus erythematosus (SLE) in a phase Ib clinical trial[30]. Despite the importance of the ICOS/ICOS-L interaction in modulating many aspects of adaptive immunity and the growing evidence of the benefits of targeting this immune complex for therapeutic interventions, the molecular details of how the extracellular domains of this receptor/ligand pair interact and how leading therapeutic antibodies recognize their targets remain elusive.

Here, we report the crystal structure of the co-complex between human ICOS and its ligand ICOS-L at 3.3 Å resolution, which reveals the molecular details of immune receptor

specificity. Furthermore, we describe the structural basis of the interactions of two therapeutic antibodies with ICOS and ICOS-L, respectively. Together, our structural characterizations uncover the molecular blueprints of the ICOS/ICOS-L interaction and a detailed view of molecular mimicry achieved by therapeutic antibodies to target the ICOS/ICOS-L signaling axis in immunotherapy.

## Results

**Structure of the ICOS/ICOS-L complex**. We determined the molecular basis for the co-stimulatory signal provided by ICOS (Fig. 1a) by solving the three-dimensional structure formed by the extracellular domains of ICOS (residues 21–129) and its ligand ICOS-L (residues 19–248) (Fig. 1b) at 3.3 Å resolution by x-ray crystallography (Table 1). Truncation of ICOS at residue 129 and therefore before C136, abolished the formation of disulfide-linked homodimers and was required to obtain well-diffracting crystals. The crystal structure reveals that ICOS adopts the predicted single immunoglobulin (Ig) variable (V-type) domain architecture. ICOS-L is organized in two distinct domains: the most apical domain (D1) adopts a V-type fold, whereas the membrane-proximal domain (D2) adopts a C1-type fold (Fig. 1c).

ICOS/ICOS-L interact in a 1:1 receptor-ligand stoichiometry. The main binding interface is formed by the FDPPPFK motif (amino acids 114–120) located in the ICOS FG loop, which interacts with residues from strands C and C' and loops CC' and C'D of ICOS-L. These residues form a network of interactions that include H-bonds and aromatic stacking (Fig. 1d, top). Single alanine substitution of residues Q50, F114, and F119 drastically impacted the binding of ICOS to ICOS-L to almost undetectable levels (Fig. 1e), confirming the critical role of this interface for receptor binding as previously reported[31].

ICOS contains three putative N-linked glycosylation sites. Clear electron density was observed in the co-complex crystal structure for two N-acetyl glucosamine (GlcNAc) and three mannose residues of the N-linked glycan at position N110. Unexpectedly, ICOS-L residues F122, Q123, and E124 form H-bonds with this ICOS glycan, which buries ~300 Å$^2$ of surface area on ICOS-L (Fig. 1d, bottom). Site-directed mutagenesis to knock out this N-linked glycosylation site (N110Q) in ICOS led to a faster on-rate and a slightly slower off-rate resulting in a 4.3-fold improvement in binding affinity to ICOS-L (Fig. 1f). This result suggests that the ICOS N110 glycan sterically gates ICOS-L binding and that the buried surface area on ICOS-L represents mere accommodation of the otherwise encumbering ICOS glycan at the binding interface.

**Comparison of ICOS and ICOS-L to other CD28 and B7 family members**. Superposition of the IgV domains of ICOS-L, B7-1, and programmed cell death ligand-1 (PDL-1) revealed that the angle of approach of ICOS to its ligand differs considerably from that observed in related family members (Fig. 2a). In the CTLA-4/B7 and PD1/PD-L1 complexes, the receptors cross the IgV domains of their cognate ligands at ~110° and 90°, respectively. However, in the ICOS/ICOS-L complex, the IgV domains cross at 150° (Fig. 2b).

These differences in disposition between the complexes are due to their distinct receptor-ligand binding interfaces. Despite low sequence identity (~20%), the overall structure of ICOS and ICOS-L share many similarities with members of their respective families (Fig. 3). Comparison of ICOS with CTLA-4 and CD28 reveal some degree of structural homology with a root mean square deviation (r.m.s.d) of backbone atoms of 3.2 and 4.5 Å, respectively, and a lower structural similarity with PD1 (r.m.s.d of 5.9 Å). Notably, the FDPPPF sequence in ICOS is analogous and

remarkably structurally conserved to the invariant motif MYPPPY present in the IgV domains of CD28[32] and CTLA-4[33,34] (backbone r.m.s.d of 0.45 and 0.39 Å, respectively). The side chains of residues flanking the three central prolines are similarly oriented in all structures as a result of the cis-trans-cis arrangement of prolines. In contrast, the analogue motif LAPKAQ in PD1 is not conserved either in sequence or in structure (Supplementary Fig. 1). Overall these motifs contribute

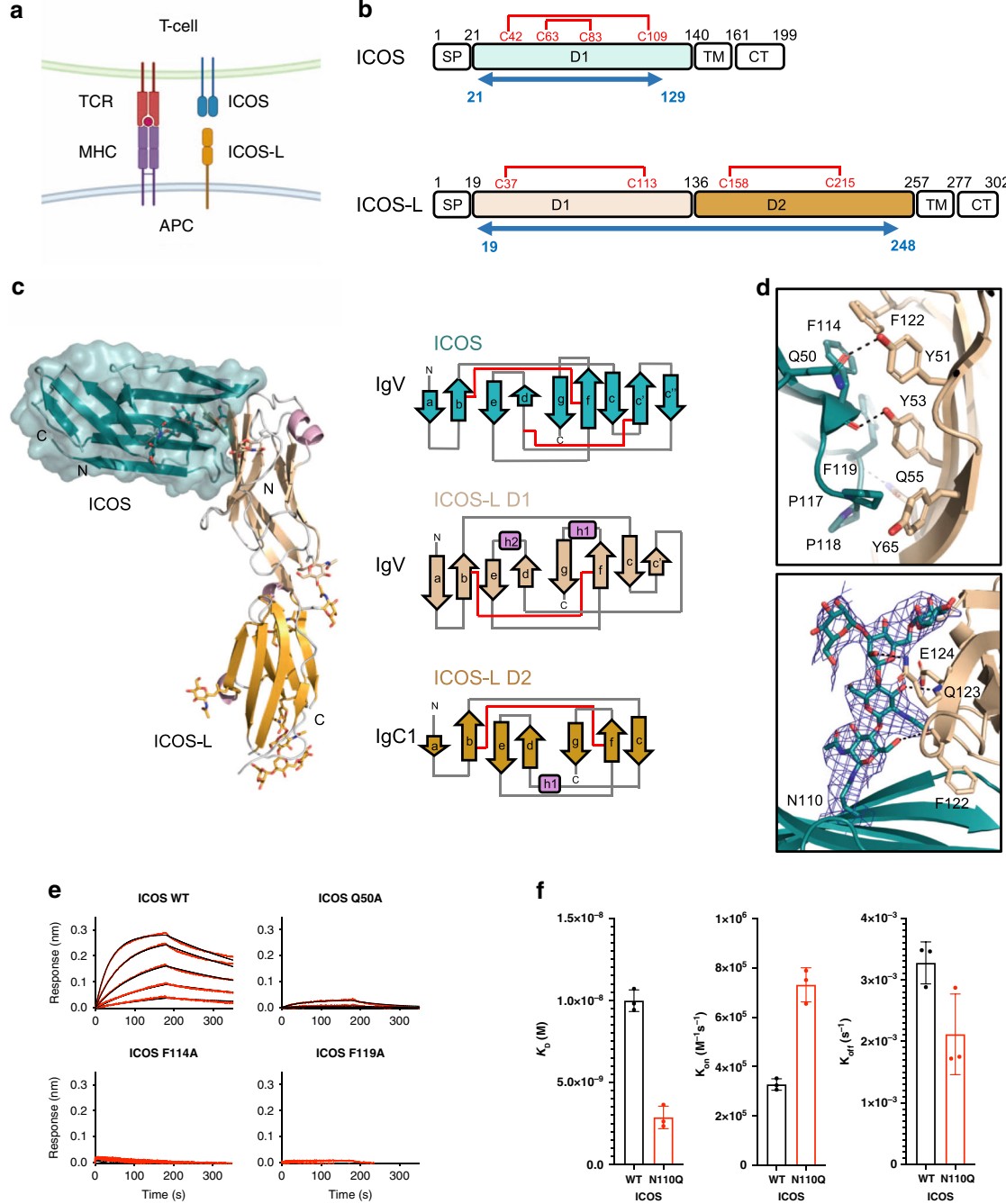

**Fig. 1 Three-dimensional structure of the human ICOS/ICOS-L complex. a** Schematic representation of the ICOS/ICOS-L co-stimulatory interaction. Antigen-presenting cell (APC), T-cell receptor (TCR), Major Histocompatibility Complex (MHC), Inducible Co-stimulator (ICOS), Inducible Co-stimulator ligand (ICOS-L). **b** Domain organization of ICOS and ICOS-L; signal peptide (SP), extracellular domains (D1-D2), transmembrane (TM) domain, and cytoplasmic tail (CT). Disulfide bonding pattern is shown with red lines. Blue arrows indicate the length of the crystalized ectodomains. **c** Secondary structure cartoon representation of the side view of the ICOS/ICOS-L complex. ICOS (deep teal, transparent surface) adopts a V-type Ig fold while ICOS-L adopts a V-type Ig fold (wheat) followed by a C1-type Ig domain fold (orange). The helices are shown in light pink. **d** Detailed view of the receptor-ligand interface. Top: the $^{114}$FDPPF$^{119}$ motif in the FG loop of ICOS forms key hydrophobic interactions with aromatic residues from ICOS-L shown in stick representation. Hydrogen bonds between ICOS and ICOS-L are shown as black dashed lines. Bottom: The blue mesh around the N110 glycan depicted as sticks represent a composite omit electron density map contoured at 1.0 sigma. Colors are as in (**c**). **e** Kinetic binding curves of ICOS-L to ICOS WT and ICOS mutants. Mutation of residues Q50, F114, and F119 abolished binding in BLI. **f** Comparison of the binding affinity ($K_D$), association (on), and dissociation ($k_{off}$) rates of ICOS-L to WT ICOS and to ICOS N110 glycan knock-out mutant (N110Q). The mean values and standard deviation of three biological replicates are shown.

**Table 1 Data collection and refinement statistics.**

| | ICOS/ICOSL/VNAR | ICOSL/Prezalumab/VNAR | ICOS/STIM003/anti-kappa $V_H$H |
|---|---|---|---|
| Data collection | | | |
| Wavelength (Å) | 1.00000 | 1.03316 | 0.97934 |
| Space group | P4(1)2(1)2 | P2(1) | C2 |
| Cell dimensions | | | |
| a,b,c (Å) | 104.0, 104.0, 123.1 | 68.5, 152, 86.7 | 171.8, 49.0, 91.8 |
| $\alpha, \beta, \gamma$ (°) | 90, 90, 90 | 90, 104.1, 90 | 90, 101.9, 90 |
| Resolution (Å) | 40–3.30 (3.40–3.30) | 40–3.15 (3.25–3.15) | 40–2.38 (2.48–2.38) |
| No. molecules in ASU | 1 | 2 | 1 |
| No. total reflections | 128,986 (4688) | 209,889 (9398) | 201,759 (20,592) |
| No. unique reflections | 10,227 (406) | 29,746 (1378) | 30,150 (3223) |
| Multiplicity | 11.8 (12.2) | 7.0 (7.0) | 6.6 (6.0) |
| Rmerge (%) | 26.9 (78.6) | 16.8 (67.0) | 10.9 (60.8) |
| Rpim (%) | 7.9 (23.0) | 6.8 (27.4) | 4.5 (25.8) |
| $<I/\sigma\ I>$ | 10.0 (1.7) | 9.4 (1.7) | 13.3 (2.1) |
| $CC_{1/2}$ | 99.3 (62.8) | 99.4 (76.2) | 99.7 (73.2) |
| Completeness (%) | 95.7 (97.9) | 99.9 (100) | 99.2 (94.0) |
| Refinement | | | |
| Non-hydrogen atoms | 3513 | 11,774 | 5245 |
| Macromolecule | 3301 | 11,580 | 5072 |
| Solvent | – | – | 153 |
| $R_{work}/R_{free}$ | 0.224/0.267 | 0.201/0.256 | 0.203/0.260 |
| Rms desviations | | | |
| Bond lenghts (Å) | 0.01 | 0.004 | 0.004 |
| Bond angles (°) | 1.14 | 0.81 | 0.72 |
| Ramachandran plot | | | |
| Favored regions (%) | 98.1 | 95.2 | 97.7 |
| Allowed regions (%) | 1.4 | 4.7 | 2.3 |
| B-factors (Å$^2$) | | | |
| Wilson B-factor | 86.6 | 76.7 | 45.7 |
| Average B-factors | 85.8 | 81.3 | 49.6 |
| Average macromolecule | 83.9 | 80.8 | 49.6 |
| Average solvent | – | – | 44.6 |

400 Å$^2$ (64%), 360 Å$^2$ (55%), 480 Å$^2$ (71%), and 230 Å$^2$ (28%) of surface area to the binding interface of ICOS/ICOS-L, CTLA-4/B7-1, CTLA-4/B7-2, and PD1/PDL-1, respectively.

A unique feature of ICOS, however, is its unusually long strand C' that protrudes from the surface of the ectodomain leading to the formation of a ligand interaction not yet observed in any of the previously solved CD28/B7 family complexes. Specifically, binding of residues $_{66}$TKTKGS$_{71}$ located in the ICOS elongated strand C' to ICOS-L buries 130 Å$^2$ of surface area, accounting for ~20% of the total BSA. Hence, although the FDPPPF motif constitutes the core of the ICOS/ICOS-L interface, residues TKTKGS in strand C' together with residues Q50, K52 in strand C contribute to the binding specificity of ICOS (Fig. 3a).

Alignment of the B7 family proteins with ICOS-L show high structural homology, with backbone r.m.s.d. ranging from 1.1 to 1.6 Å. Unsurprisingly, the primary structural difference observed in ICOS-L with family members is found in the conformation of loop C'D, which contacts strand C' of ICOS (Fig. 3b). As a result, the binding interface of ICOS-L is shifted toward the edge of the AGFCC' sheet and involves residues from strands F (L114, L116) and C (Y51, Y53, Q55), as well as loops CC' (K60, V62), C'D (I67, Q69, N75) and FG (Q118, G121, F122). Consistently, mutations in any of these residues were previously reported to result in a substantial loss of receptor binding[35].

**Oligomeric engagement of the ICOS/ICOS-L immune complex.** Due to the ability of ICOS to form disulfide-linked homo-dimers[3] and the importance of surface oligomerization to trigger its co-stimulatory signal, we measured binding affinity and binding avidity of the ICOS/ICOS-L complex by biolayer interferometry (BLI) (Fig. 4). Binding of the ICOS-L ectodomain monomer to the ICOS ectodomain immobilized as an array on the biosensors showed a relatively low binding affinity ($K_D = 722$ nM) and very rapid off-rates ($K_{off} = 1.6 \times 10^{-1}$ s$^{-1}$) (Fig. 4, top panel and Supplementary Table 1). In contrast, by flipping the orientation of the interaction and measuring binding avidity of ICOS disulfide-linked homodimers to immobilized ICOS-L arrayed on the biosensor, two orders of magnitude difference in the apparent binding affinity constant ($K_D = 10$ nM) was obtained (Fig. 4, bottom panel and Supplementary Table 1). Such high avidity might reflect clustering and oligomerization of ligands and receptors at the cell surface upon T-cell activation and is in agreement with the propensity of ICOS and ICOS-L to form homodimers at the cell membrane[3,35].

Our molecular data provide partial insight into the oligomeric assembly of ICOS and ICOS-L. Indeed, in the absence of an interchain disulfide bond, ICOS exists as a monomer in solution (Supplementary Fig. 2a). In our crystal structure where the ICOS interchain disulfide bond has been truncated, we observe an ICOS/ICOS-L interface that buries ~420 Å$^2$ of BSA (Supplementary Fig. 2b). This interaction could be indicative of an ICOS dimeric interface present at the cell membrane; however, in this crystallographic arrangement the ICOS protomers are oriented in a head to tail fashion with the C-terminal tails (where the cysteine residues would mediate putative disulfide bonds) pointing away from each other (Supplementary Fig. 2c). As such, in the absence of functional data supporting this dimeric interface, we propose that the antiparallel ICOS dimer arrangement observed in the crystal lattice is likely an artefact of crystal packing. Altogether,

our data suggest that ICOS dimerization at the cell surface probably involves a limited dimeric interface of the ectodomain and that the ICOS ectodomain is mainly linked through the

disulfide bond at the C-terminal tail near the transmembrane region (Supplementary Fig. 2d).

**Therapeutic antibodies can compete with the ICOS/ICOS-L interaction.** Monoclonal antibodies STIM003 and prezalumab target ICOS and ICOS-L, respectively and are currently under clinical evaluation. Using BLI, we confirmed that binding of these antibodies to their respective ligands prevents the formation of the ICOS/ICOS-L immune complex (Fig. 5a). To elucidate the molecular basis of ligand blockade by the antibodies, we solved the crystal structures of ICOS/STIM003 Fab and ICOS-L/prezalumab Fab complexes at 2.38 and 3.15 Å resolution, respectively (Table 1).

Superposition of the ICOS structure, when crystallized in complex with its natural ligand and in complex with STIM003, showed high structural similarity (backbone r.m.s.d of 0.90 Å). The binding interface of ICOS in the antibody complex was strikingly similar to the one when bound to its natural ligand (Fig. 5b). In both cases, binding is concentrated in the FG loop of ICOS. However, the total buried surface area of ICOS is slightly higher for the STIM003 interaction in comparison to the interaction with ICOS-L (800 to 620 Å$^2$, respectively), due to additional interactions of the antibody with the C'C" loop and the C" strand. Despite the high similarity of the contacted ICOS surface, ICOS-L and STIM003 approach the receptor with different angles of approach (Fig. 5b). Consequently, the ICOS N110 glycan minimally interferes with binding to STIM003 (Supplementary Fig. 3 and Supplementary Table 1).

The crystal structure of the ICOS-L/prezalumab Fab complex was obtained using a shark Variable New Antigen Receptor (VNAR) Single Domain[36] as a crystallography chaperone (Supplementary Fig. 4a). The crystal structure revealed that ICOS and prezalumab bind ICOS-L with a similar angle of approach, and that the ICOS IgV domain is in a highly similar conformation in both complexes (backbone r.m.s.d of 1.0 Å). Both molecules interact with an overlapping surface on ICOS-L; however, prezalumab makes additional contacts with strand F and loop C'D of ICOS-L upon binding, leading to a higher BSA compared to the natural complex (900 Å$^2$ vs. 580 Å$^2$, respectively) (Fig. 5c). As a result of the extended interface, prezalumab binding is proximal to N70 (Fig. 6a), a predicted N-linked glycosylation site. However, the crystal structure reveals almost no interaction with N70 or the N-linked glycan (BSA of 10 and 60 Å$^2$, respectively) therefore suggesting that binding of prezalumab to ICOS-L is largely independent of N-glycosylation. Notably, the angle between the ICOS-L IgV and IgC domains in the ICOS/ICOS-L and ICOS-L/prezalumab Fab crystal structures differs by ~14°, indicating flexibility that can be attributed to a largely flexible linker and minimal interactions between the two ICOS-L Ig domains (Supplementary Fig. 4b).

**Antibody mimicry of the ICOS/ICOS-L interaction.** Remarkably, residues at the core of the ICOS/ICOS-L interaction are also central to the antibody complexes (ICOS/STIM003 and ICOS-L/prezalumab) (Fig. 6a). Prezalumab uses the heavy chain complementary determining region (CDR) 3 (HCDR3) to mimic the central ICOS FG loop that engages with ICOS-L. On the other hand, HCDR3, LCDR1, and LCDR2 residues of STIM003 resemble the hydrophobic residues of the ICOS-L front sheet that drive the interaction with ICOS. A key feature of the antibody-antigen interactions is that in addition to central hydrophobic contacts that resemble the natural ligands, additional hydrogen bonds and salt bridges form in the periphery (Fig. 6b). Indeed, only five hydrogen bonds are formed at the binding interface between ICOS and ICOS-L. In contrast, ten hydrogen bonds and

**Fig. 2 Angle of ligand-receptor interactions for CTLA-4/B7 family members. a** Top view cartoon representation of ICOS/ICOS-L and previously solved CTLA-4/B7-1[33] and PD1/PDL-1[69] complexes. The orientation of the complexes are based on the structural alignment of the ligands (different shades of brown). Receptors are shown as shades of green. Structural features are labeled to help with orientation. **b** Angle of approach of the three receptors to its ligands calculated using Pymol[70].

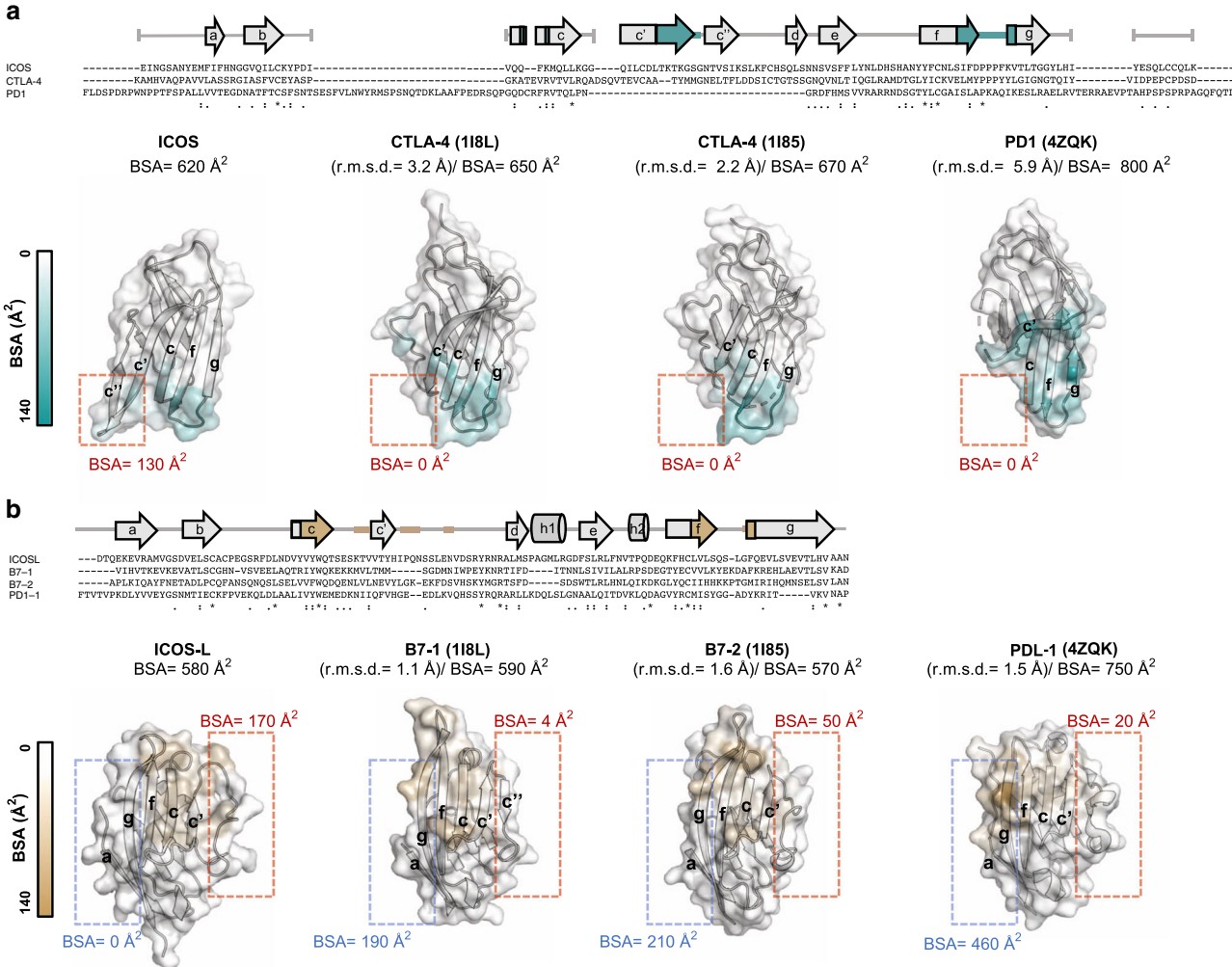

**Fig. 3 Specificity of the ICOS and ICOS-L interaction.** Sequence alignment of (**a**), ICOS with CTLA-4 and PD1 and **b**, ICOS-L with B7-1, B7-2, and PDL-1. Invariant residues (asterisk), and residues with highly (colon) and weakly similar properties (dot) are indicated. Strands and α helixes are denoted by arrows and cylinders, respectively. Secondary structure cartoon representation of (**a**), the V-type domains of ICOS in comparison with CTLA-4 (PDB IDs: 1I8L and 1I85) and PD1 (PDB ID: 4ZQK) and (**b**), ICOS-L in comparison with B7-1 (PDB ID: 1I8L), B7-2 (PDB ID: 1I85) and PDL-1 (PDB ID: 4ZQK). Backbone r.m.s.d. between ICOS and ICOS-L and structures of other family members is shown in parentheses and were calculated in Pymol[70]. Residues are shown as surface and colored according to their BSA. Areas with higher and lower BSA in ICOS and ICOS-L with respect to the other family members are boxed in red and blue, respectively. The total BSA of each ectodomain is indicated.

one salt bridge (ICOS/STIM003), and seven hydrogen bonds (ICOS-L/prezalumab) are part of the antibody-antigen binding interfaces. These additional interactions are concentrated in the enlarged areas of the antibody interfaces that is solvent accessible in the ICOS/ICOS-L complex (Fig. 5) and are presumably responsible for the higher binding affinity of the antibodies compared to the natural ligands (Fig. 6c and Supplementary Table 1). Noteworthy, the mimicry of these two antibodies compared to their cognate receptors is higher than any other therapeutic antibodies targeting receptors/ligands in the same family for which the three-dimensional structure has been reported (Fig. 6d and Supplementary Fig. 5).

## Discussion

T cell activation is mediated by the coordinated interplay of a complex network of transmembrane co-stimulatory and co-inhibitory receptor/ligand pairs. When dysregulated, T cell responses can result in tissue damage leading to autoimmunity or cancer. Thus, modulation of these molecular complexes to repress or enhance immune responses is a powerful strategy in

immunotherapy. Important examples are provided by the remarkable improvements in disease outcome obtained with antibodies targeting CTLA-4 and PD1[37–40]. ICOS, a member of this family, has emerged as a promising target for immunotherapy[24–30] because of its central role in the T/B-cell co-signaling pathway[3] associated with adaptive and innate immunity[21,41].

Our crystal structure of the ICOS/ICOS-L complex revealed that ICOS adopts a predicted[35] overall Ig-fold structure similar to CTLA-4 and CD28[42]. These three surface receptors utilize a central PPP motif disposed in a high-energy cis-trans-cis conformation that is flanked by aromatic residues to engage their cognate ligands[32–34]. However, we show that unlike CTLA-4/CD28, ICOS utilizes a second set of residues within its CC' loop to contribute a considerable fraction of the contacts likely responsible for its binding specificity to ICOS-L. Interestingly, identification of this second binding interface offers an opportunity to guide loop grafting strategies for the design of soluble ligands with cross-reactivity between receptors. In line with this idea, simultaneous blockade of the co-stimulatory receptors ICOS and CD28 have been effective in prolonging cardiac allograft

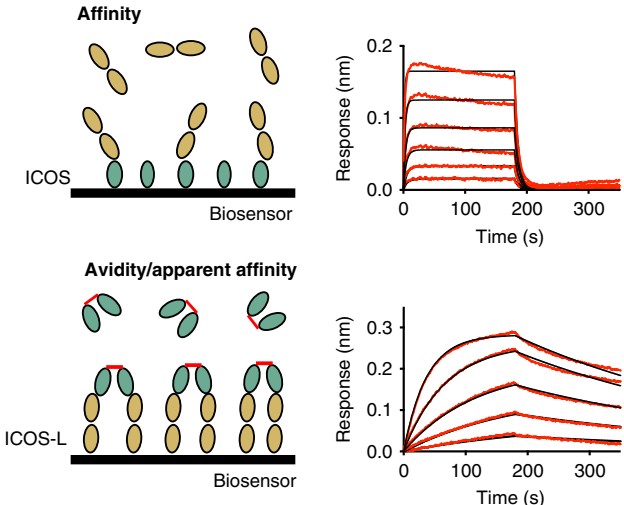

**Fig. 4 Binding of recombinant ICOS and ICOS-L ectodomains.** Schematic experimental set-up (left) and kinetic binding curves (right) for immobilized ICOS and ICOS-L as analyte (top) and inverted system (bottom) measured by BLI.

survival in rats[28]. Therefore, we speculate that such cross-reactive and biparatopic molecules could show beneficial therapeutic properties for several disease indications, including allograft transplantation, autoimmune diseases, and cancer.

Monovalent binding of ICOS to ICOS-L showed low affinity and fast binding kinetics, while bivalent binding resulted in a two-orders of magnitude stronger binding kinetics. Rapid off-rates might be important to allow Tfh-cell motility into the B cell zone in the GC through a dynamic formation and disruption of T cell-APC contacts facilitating fast antigen scanning. This process is governed by ICOS/ICOS-L interaction in a TCR-independent manner[43]. However, efficient co-stimulatory signaling relies on avidity for the formation of oligomeric structures with high stability at the immunological synapse. Interestingly, this dependence on binding avidity observed for ICOS has been previously reported for CTLA-4 homodimers, but not for CD28 homo-dimers[44]. Indeed, CTLA-4 and B7-1 can form a zipper-like oligomerization of disulfide-linked CTLA-4 homodimers and B7-1 homodimers[33]. On the contrary, CD28 homodimers are incompatible with the simultaneous binding of two B7-1 molecules due to clashes of the membrane-proximal domain of the ligand[42]. Our crystal structure of the ICOS/ICOS-L monomeric complex did not allow to define a biologically relevant dimer interface for the ICOS ectodomain; indeed, a mutation removing the interchain disulfide bond in ICOS was required to obtain well-diffracting crystals, resulting in the ICOS ectodomain being monomeric in solution and limited information could be derived from the crystal packing interfaces. Similarly, monomeric forms of CTLA-4 in solution have been reported in the absence of the disulfide linkage[32,45]. This observation, together with the capacity of ICOS and CTLA-4 to simultaneously bind two ligand molecules, suggest that ICOS and ICOS-L may leverage a periodic arrangement of receptor-ligand complexes at the T-cell-APC interface similar to the one reported for the CTLA-4/B7-1 complex. However, future studies will be required to precisely define the oligomeric assembly of these molecules at the immune synapse.

A notable feature uncovered by our ICOS/ICOS-L co-complex structure is the presence of ICOS N-linked glycan N110 at the binding interface. The N110Q mutation removing this N-linked glycan resulted in a 4.3-fold improvement in binding affinity of ICOS to ICOS-L. Such a role of N-linked glycosylation modulating receptor binding was also reported in previous studies

where deglycosylation of CD28 was found to enhance binding to CD80 on APC[46], and a hypoglycosylated form of B7-2 showed reduced binding to CD28 and CTLA-4[47]. Importantly, abnormal glycosylation of cell surface proteins can influence signaling pathways implicated in cell survival and growth-promoting several disorders, including cancer[48]. For example, a recent study reported that N-glycan modification of PD-L1 on triple-negative breast cancer cells was essential for PD-1 interaction and therefore T cell exhaustion[49]. It remains to be determined how ICOS glycosylation is impacted in dysregulated cells, and how such modifications in the presence and composition of the N110 glycan may impact disease progression. Nonetheless, our findings contribute further evidence of post-translation modifications, and particularly glycobiology in modulating binding thresholds, in this case relevant for T cell activation.

Therapeutic antibodies with an antagonistic mode of action often compete with natural ligands[50], and their efficacy is linked to the extent of epitope overlap with the natural ligand footprint. Yet, steric overlap is rarely achieved by exact mimicry of the native molecular interactions. Strikingly, our crystallographic studies revealed that two therapeutic antibodies, STIM003 and prezalumab, contact almost all residues involved in the ICOS/ICOS-L immune complex interaction. Comparison to other therapeutic antibody-ligand structures in the checkpoint blockade family (PD1, PDL-1, and CTLA-4) revealed that the level of mimicry observed for STIM003 and prezalumab is remarkable. Accordingly, our structural data suggest that the reported antagonistic effect of prezalumab[51] is likely attributed to efficient outcompeting ICOS for binding to ICOS-L, in addition to possible steric hindrance that block receptor-ligand clustering at the membrane surface. The level of mimicry achieved by the ICOS and ICOS-L therapeutic antibodies described here approaches some of the previous reports in the field of infectious diseases where antibodies against viral envelope proteins precisely target specific conserved receptor-interacting residues to avoid viral escape[52–55]. The molecular principles derived from our structures will continue to inform de novo structure-based strategies for the design of biologics that showcase natural mimicry, as also recently exemplified by the design of selective mimics of IL-2 and IL-15[56].

A detailed analysis of the binding interface of ICOS/STIM003 and ICOS-L/prezalumab also revealed a slightly larger footprint compared to the binding interface of the receptor-ligand pair, and the presence of additional hydrogen bonds and salt bridges in the periphery of the core interaction. In the BioMuta sequence database of cancer patients[57], eight single nucleotide variations (SNVs) compiled in the ICOS sequence (S76P, G70R, N73Y, K78N, F114V, P116S, P116H, and P117A) and two in the ICOS-L sequence (Y51C and Y65H) were identified as antibody-contact residues (Supplementary Fig. 6a). How these SNVs will impact therapeutic responses is an area of future investigation now possible from the precise delineation of these antibody epitopes.

Together, our data provide critical knowledge to better understand the molecular basis of the APC/T cell interaction, and the atomic blueprints for the design of next-generation biologics to modulate the ICOS/ICOS-L therapeutic axis.

## Methods

**ICOS and ICOS-L expression and purification**. The ectodomains of human ICOS (UniprotKB Q9Y6W8) (residues 21–138) and ICOS-L (UniprotKB O75144) (residue 19–248) followed by a tobacco etch virus (TEV) cleavage site were fused to a monomeric variant of Venus[58] to promote expression of the glycoproteins. Genes were synthesized at GeneArt (Life Technologies) and cloned into the pHLsec expression vector containing a C-terminal $His_{6x}$ tag for downstream purification. Proteins were expressed in HEK 293F cells (ThermoFisher Scientific) following standard protocols[59]: about 200 mL of cells were seeded at a density of $0.8 \times 10^6$ cells/mL and incubated with 125 rpm oscillation at 37 °C, 8% $CO_2$, and 70%

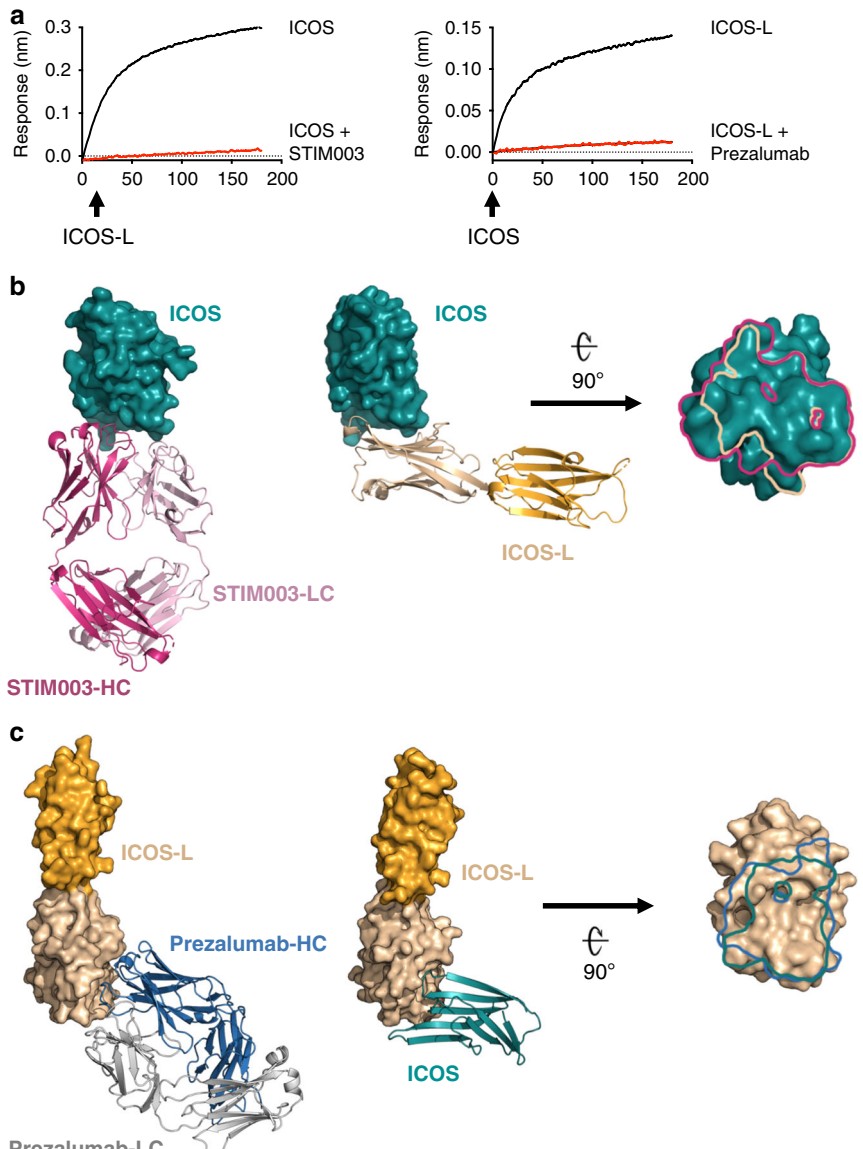

**Fig. 5 Recognition of ICOS and ICOS-L by therapeutic antibodies. a** Antibody binding competition of ICOS-L (immobilized) to ICOS or ICOS + STIM003 Fab (left); and ICOS (immobilized) to ICOS-L or ICOS-L + Prezalumab Fab (right). Comparison between the structures of the ICOS/ICOS-L complex and **b** the ICOS/STIM003 complex and **c** the ICOS-L/prezalumab complex. The light and the heavy chains of STIM003 are colored in light and dark pink, respectively while the prezalumab light and the heavy chains are colored in gray and blue, respectively. ICOS (deep teal) and ICOS-L (wheat) are rotated 90° about a horizontal axis to reveal the binding surface of the antibodies. Epitope traces of STIM003 (pink) and ICOS-L (wheat) are depicted on the surface of ICOS, and prezalumab (blue) and ICOS (deep teal) on the surface of ICOS-L.

humidity in a Multitron Pro shaker (Infors HT). Twenty-four hours after seeding, cells were transiently transfected using 50 μg of filtered DNA preincubated for 10 min at room temperature (RT) with the transfection reagent FectoPRO (Poly-plus Transfections) in a 1:1 ratio. Cell suspensions were harvested by centrifugation at $5000 \times g$ for 15 min after 6–7 days and the supernatants were passed through a HisTrap Ni-NTA column (GE Healthcare) at 4 ml min$^{-1}$. After washing the column with 20 mM Tris pH 9.0, 150 mM NaCl, 5 mM imidazole, ICOS-Venus, and ICOS-L-Venus were eluted with an increasing gradient of imidazole (up to 500 mM). Fractions containing protein were pooled and digested for 1 h at RT with the TEV protease. Digested protein was recovered in the flow through of a second HisTrap Ni-NTA column, concentrated and loaded onto a Superdex 200 Increase size exclusion column (GE Heathcare) in 20 mM Tris pH 9.0, 150 mM NaCl buffer.

**Expression and purification of ICOS and ICOS-L complexes**. For structure determination of the ICOS/ICOS-L complex, a shark Variable New Antigen Receptor (VNAR) Single Domain targeting the constant domain of ICOS-L was used[36]. The ternary complex was produced by co-transfecting the DNA encoding ICOS (residues 21–129): ICOS-L (residues 19–248):VNAR in a 3:1:4 ratio.

Similarly, the ICOS-L/prezalumab complex was produced in the presence of VNAR by co-transfecting ICOS-L:Fab$_{HC}$: Fab$_{LC}$:VNAR in a 2:2:1:4 ratio. The ICOS/ STIM003 complex was obtained by co-transfection of ICOS C136AC137A N23Q (residues 21–138):Fab$_{HC}$:Fab$_{LC}$ in a 2:2:1 ratio. Cys mutations were required to increase sample homogeneity and obtain well-diffracting crystals. The DNA ratios used in each transfection was selected based on the expression yields of the indi-vidual proteins with the aim to achieve similar expression levels between them when co-transfecting. In order to obtain samples of homogeneous glycan com-position that would allow downstream processing and efficient crystal packing, the three complexes were expressed in HEK 293S cells (Gnt I$^{-/-}$). After harvesting the cells, the supernatant was loaded onto a HisTrap Ni-NTA column and the complex eluted with a gradient of imidazole. Upon buffer exchange to remove the imidazole, Venus and glycans were cleaved by incubating with TEV and EndoH, respectively for 1 h at 37 °C. The digested complex was collected from the flow through of a second HisTrap Ni-NTA column, concentrated and loaded onto a Superdex 200 Increase size exclusion column (GE Heathcare) in 20 mM Tris pH 9.0, 150 mM NaCl buffer. In addition, for the ICOS/ICOS-L complex, the fractions containing protein were buffer exchanged to 20 mM Tris pH 9.0, loaded on a MonoQ ion exchange column and eluted with a 0–50% linear gradient of 1 M potassium chloride in 20 mM Tris pH 9.0 buffer.

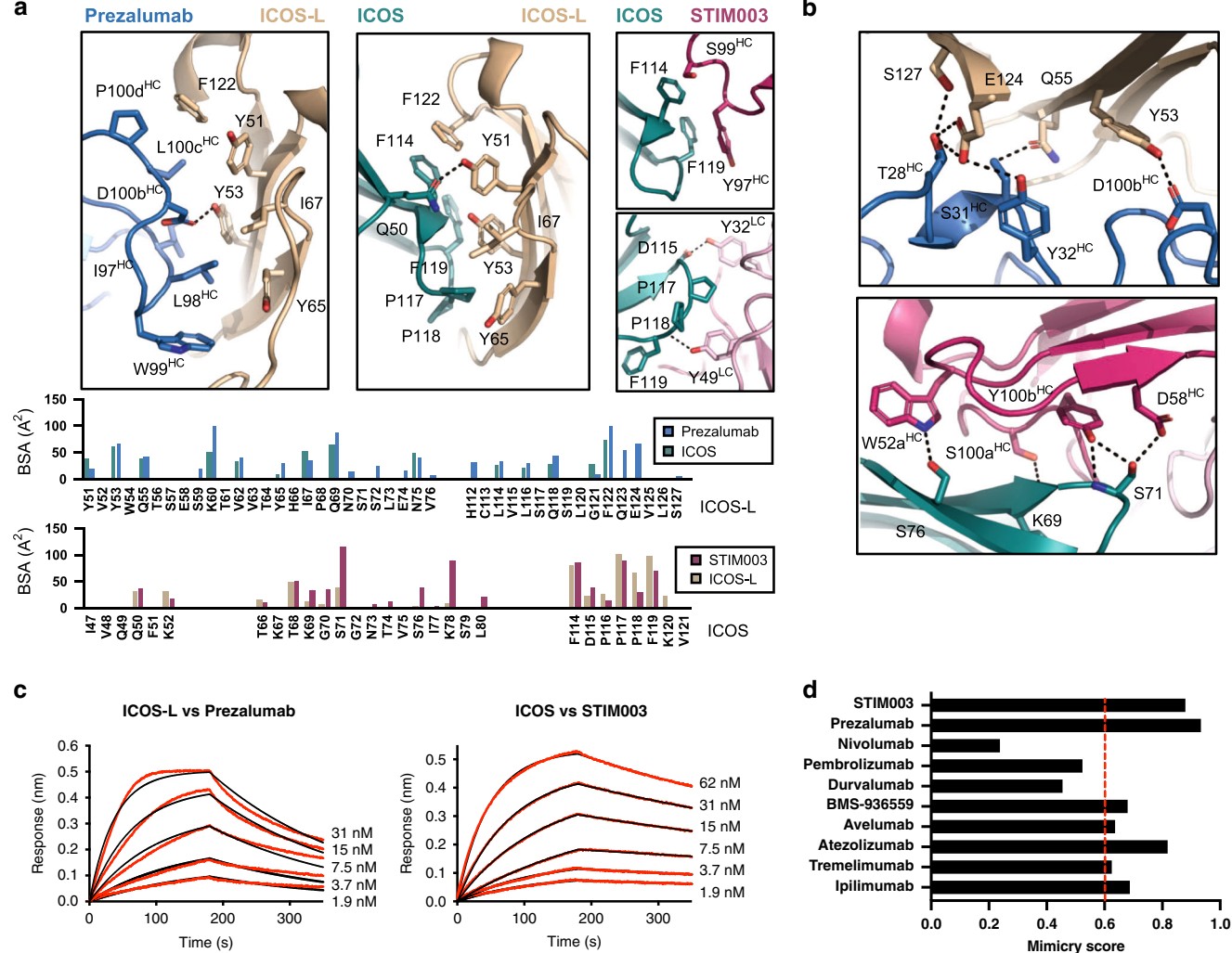

**Fig. 6 Remarkable receptor mimicry by antibodies targeting ICOS/ICOS-L. a** Antibody mimicry of the main hydrophobic/aromatic interface of the ICOS/ICOS-L complex by STIM003 and prezalumab. HCDR3 of prezalumab mimics the FG loop of ICOS to mediate interaction with ICOS-L (left panel). STIM003 interactions with the FG loop of ICOS are mediated by residues from HCDR3, LCDR1 and LCDR2 (right panels). Critical residues for binding are shown in sticks and the BSA of these and the rest of residues involved in binding are plotted (bottom). The BSA of the ICOS/ICOS-L interaction is included for direct comparison. **b** Additional H-bonds (black dashes) between the antibodies and ICOS and ICOS-L in areas outside the core interface of the receptor/ligand complex. **c** Kinetics of prezalumab and STIM003 binding to ICOS-L and ICOS, respectively. **d** Comparison of binding mimicry by therapeutic antibodies targeting the CD28/B7 superfamily calculated as the fraction of residues with >1 Å² of BSA of the receptor/ligand pair that are contacted by antibody binding as detailed in Supplementary Fig. 5. Color coding is as in Fig. 4.

**Fab expression and purification**. The Fab heavy and light chains of STIM003 and prezalumab were cloned into custom pcDNA3.4 expression vectors at GeneArt (Life Technologies). Fabs were transiently expressed in 200 mL HEK 293F cells (ThermoFisher Scientific) by co-transfecting 90 μg of the LC and the HC in a 1:2 ratio with FectoPRO (Polyplus Transfections). Purification of the Fabs was performed using a KappaSelect affinity column (GE Healthcare) and 100 mM glycine pH 2.2 as the elution buffer. Eluted fractions were immediately neutralized with 1 M Tris-HCl pH 9.0 and further purified using a MonoS ion exchange column and a Superdex 200 Increase size exclusion column (GE Healthcare).

**Crystallization and X-ray data collection**. Purified complexes were concentrated, mixed 1:1 with mother liquor and set up in sitting drop vapor diffusion crystallization experiments. The ICOS/ICOS-L/VNAR complex was concentrated to 10 mg/mL and crystals grew in 0.1 M HEPES, pH 7.0, and 30% (v/v) Jeffamine ED-2001 and were cryoprotected with 15% (v/v) glycerol. X-ray diffraction data were collected at the 17-ID-B synchrotron beamline at the Advanced Photon Source (APS) at the Argonne National Laboratory. The ICOS-L/prezalumab/VNAR complex was concentrated to 12 mg/mL and grown in 0.2 M di-ammonium tartrate and 20% (w/v) PEG 3350 and cryoprotected with 10% (v/v) glycerol. X-ray diffraction data were collected at the 23-ID-B synchrotron beamline at APS. To determine the structure of the ICOS/STIM003 complex, the purified complex was incubated with a variable heavy-chain (V$_H$H) domain specific for the human kappa

light chain as a crystallization chaperone[60] in a 1:5 ratio for 30 min at RT followed by gel filtration chromatography (Superdex 200 Increase size exclusion column, GE Healthcare) in 20 mM Tris pH 9.0, 150 mM NaCl buffer. The purified sample was concentrated to 4 mg/mL and crystals were obtained in 0.2 M di-sodium hydrogen phosphate and 20% (w/v) polyethylene glycol 3350 and cryoprotected with 10% (v/v) glycerol. X-ray diffraction data were collected at the FMX synchrotron beamline at the National Synchrotron Light Source II (NSLS-II) at Brookhaven National Laboratory (BNL). Data from the three complexes were processed using XDS[61] and the structures were solved by molecular replacement using Phaser[62]. CTLA-4[33], B7-H3[63], B7-1 D2[33], Fabs from our internal database, and VNAR type 1[64] were used as search models for ICOS, ICOS-L D1, ICOS-L D2, the Fabs, and VNAR, respectively. The refinement of the structures was carried out by iterative rounds of phenix.refine[65] and manual building in Coot[66]. Representative electron density for the three structures is shown in Supplementary Fig. 6b–d. EMBL-EBI-PDBePISA[67] was used to calculate the reported buried surface area. Access to all software was supported through SBGrid[68].

**Biolayer interferometry**. Binding affinity measurements were conducted by BLI using an Octet RED96 BLI system (Pall ForteBio) in PBS pH 7.4, 0.01% BSA, and 0.002% (v/v) Tween. Ni-NTA biosensors were used to bind His-tagged ICOS and ICOS-L proteins. A signal response of 0.8 nm was reached before transferring the loaded biosensors to wells containing a 1:2 serial dilutions of the Fabs or of the

untagged ICOS-L and ICOS, respectively. Initial concentrations of 125, 500, and 31 nM were used for ICOS (wild-type and mutants), ICOS-L and prezalumab, respectively when used as analytes. The duration of each association and dissociation steps was 180 s. The analysis was performed using the Octet software, with a 1:1 fit model. A minimum of three replicates for each binding curve was done. Supplementary Table 1 contains the averaged values and the calculated standard deviations of the kinetic measurements.

## Data availability

The crystal structures reported in this manuscript are available from the Protein Data Bank under accession codes 6X4G (ICOS/ICOS-L/VNAR complex), 6X4T (ICOS-L/Prezalumab/VNAR complex), and 7JOO (ICOS/STIM003/anti-kappa V$_H$H complex). Other data are available from the corresponding author upon reasonable request.

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

## Acknowledgements

This work was supported by the European Union's Horizon 2020 research and innovation program under Marie Sklodowska-Curie grant 790012 (E.R.), by operating grant PJT-148811 from the Canadian Institutes of Health Research (J.P.J.), the CIFAR Azrieli Global Scholar program (J.P.J.), the Ontario Early Researcher Awards program (J.P.J.), and the Canada Research Chairs program (J.P.J.). T.S. is a recipient of a Vanier Canada Graduate Scholarship. The BLI instrument was accessed at the Structural & Biophysical Core Facility, The Hospital for Sick Children, supported by the Canada Foundation for Innovation and Ontario Research Fund. X-ray diffraction experiments were performed at GM/CA@APS, which has been funded in whole or in part with federal funds from the National Cancer Institute (ACB-12002) and the National Institute of General Medical Sciences (AGM-12006). The Eiger 16M detector at GM/CA-XSD was funded by NIH grant S10 OD012289. This research used resources of the Advanced Photon Source, a U.S. Department of Energy (DOE) Office of Science user facility operated for the DOE Office of Science by Argonne National Laboratory under contract DE-AC02-06CH11357. X-ray diffraction experiments were also performed at the NSLS-II, a U.S. Department of Energy (DOE) Office of Science User Facility operated for the DOE Office of Science by BNL under Contract No. DE-SC0012704. The Life Science Biomedical Technology Research resource is primarily supported by the National Institute of Health, National Institute of General Medical Sciences (NIGMS) through a Biomedical Technology Research Resource P41 grant (P41GM111244), and by the DOE Office of Biological and Environmental Research (KP1605010).

## Author contributions

E.R. and J.P.J. conceived the research and designed the experiments; E.R., H.C., T.S., and A.S. performed experimental work. E.R. and J.P.J. analyzed the data and wrote the manuscript.

## Competing interests

The authors declare no competing interests.
