## [Peer Review File · Nature Communications]

REVIEWER COMMENTS

Reviewer #1 (Remarks to the Author):

Rujas et al detail an interesting structurally centric report on a key immunoreceptor-ligand interaction (ICOS/ICOS-L) that plays a central role in T cell stimulation. The ICOS/ICOS-L structure provides a basis for comparing against other receptor/ligand pairs within this family, and some noteworthy points of comparison and difference. Further, the manuscript details structures of two Fab fragments of clinically relevant antibodies bound to ICOS and ICOS-L, and noted a convergence of binding sites of these Fabs to that of the physiological interaction. The work is also complemented by some mutagenesis/binding studies. The work is technically excellent, and the manuscript well written with compelling figures. While the manuscript could be considered a 'straightforward' structural paper, this would overlook the significant protein chemistry expertise required to enable structural data to be obtained. As such, I only have relatively minor comments as a way of suggesting improving the paper further.

- 1) The claim in the abstract re: N110 acting as a gating element is overstated and unproven – and should be removed. Just because an impact was observed upon mutation of this site does not permit such a definitive statement to be made
- 2) Structures were actually solved with VNARs – a detail that is buried in the methods. This should be upfront and also included in the overview structural figures – as presently the authors are not actually showing the complete structure solved. The protein : protein ratios used in obtaining crystals were also very unusual – can the authors clarify why?
- 3) Low resolution structures should not have distance measurements for h-bonds/salt bridges in tables, and the BSA values should be rounded to the nearest 10.
- 4) The statement relating to the CC' loop of ICOS being responsible for binding specificity to ICOS-L will need to be watered down or backed up formally with experimental data (e.g. mutational analyses)
- 5) the ICOS-Fab structure was solved using the longer construct for ICOS. So, was this a disulfide-linked homodimer in the crystal structure, and does the monomeric/dimeric state of ICOS impact on Fab binding?

Reviewer #2 (Remarks to the Author):

The ICOS/ICOS-L pathway plays important roles in T cell co-stimulation and T follicular helper (Tfh) cell generation. In addition, this pathway has been reported to be involved in cancers and autoimmune diseases. People in the field have been waiting for the structure of ICOS and/or ICOS-L, now Julien and his colleagues reported structural characterization of the ICOS/ICOS-L immune complex and ICOS-mAb complex. The structure revealed several new discoveries, including that a central FDPPPF motif and residues within the CC' loop of ICOS are responsible for the the interaction with the ligand, N-linked glycosylation of ICOS 32 N110 is a gating element for ICOS-L binding, distinct receptor binding orientation compared to CTLA-4/B7-1 and PD-1/PD-L1. Finally, the authors showed that two therapeutic anti-ICOS mAbs bound ICOS similar to the receptor-ligand binding core interactions.

Overall, the manuscript is well-written and the structures are new and important. That being said, this Reviewer did not have a major concern. The below minor issues can be fixed to further enhance the quality of the manuscript:

1) It is well accepted that ICOS proteins are expressed as homodimer on immune cell surface. However, the structure of ICOS in this manuscript is monomer, due to the fact that only amino acid 21-129 of ICOS were used to generate proteins. This weakness should be discussed.

2) The format of all references should be unified. Some references missed page numbers.

Reviewer: Xingxing Zang

Reviewer #3 (Remarks to the Author):

Rujas et al. is really just simply a beautiful manuscript: concise yet complete, rigorously performed and interpreted, interesting, important, and beautifully illustrated. All needed information is included, the scope is pleasingly comprehensive, and the impact is manifest. As-is, the manuscript is written to be widely accessible and will hold interest for a broad swath of the community.

My recommendation: publish as-is. Well, the error on the "STIM003/ICOSdimer" interaction in Extended Table 1 should be reported as " 2 ± 3 " instead of " 2.4 ± 3.3 ", but everything else is OK.

Point-by-point response to the Reviewers' comments

Reviewer #1:

Rujas et al detail an interesting structurally centric report on a key immunoreceptor-ligand interaction (ICOS/ICOS-L) that plays a central role in T cell stimulation. The ICOS/ICOS-L structure provides a basis for comparing against other receptor/ligand pairs within this family, and some noteworthy points of comparison and difference. Further, the manuscript details structures of two Fab fragments of clinically relevant antibodies bound to ICOS and ICOS-L, and noted a convergence of binding sites of these Fabs to that of the physiological interaction. The work is also complemented by some mutagenesis/binding studies. The work is technically excellent, and the manuscript well written with compelling figures. While the manuscript could be considered a 'straightforward' structural paper, this would overlook the significant protein chemistry expertise required to enable structural data to be obtained. As such, I only have relatively minor comments as a way of suggesting improving the paper further.

We thank the Reviewer for the encouraging comments and appreciation of the manuscript.

1) The claim in the abstract: N110 acting as a gating element is overstated and unproven – and should be removed. Just because an impact was observed upon mutation of this site does not permit such a definitive statement to be made

As requested by the Reviewer, we have modified this sentence in the Abstract to more adequately describe the data reported in the manuscript: “Furthermore, our structure and binding data reveal that the ICOS N110 N-linked glycan participates in ICOS-L binding.”

2) Structures were actually solved with VNARs – a detail that is buried in the methods. This should be upfront and also included in the overview structural figures – as presently the authors are not actually showing the complete structure solved. The protein: protein ratios used in obtaining crystals were also very unusual – can the authors clarify why?

We appreciate the concern raised by the Reviewer, and agree with the suggestion. As such, the following sentence has been added in line 208 to acknowledge the VNAR as part of the ICOS-L/antibody crystal structure: “The crystal structure of the ICOS-L/prezalumab Fab complex was obtained using a shark Variable New Antigen Receptor (VNAR) Single Domain⁵⁹ as a crystallography chaperone (Supplementary Fig. 4a).”

Supplementary Fig. 4a was inserted to represent the overview of the full crystal structure, and has the following figure legend: “a) Binding of VNAR to ICOS-L. Inset: close-up view of the binding interface, which buries approximately 720 Å² of surface area on ICOS-L. Residues contributed by VNAR and ICOS-L are depicted in raspberry and orange, respectively.”

“The DNA ratios used in each transfection was selected based on the expression yields of the individual proteins with the aim to achieve similar expression levels between them when co-transfecting.” This clarifying sentence has been added to the Materials & Methods section on lines 370-372.

3) Low resolution structures should not have distance measurements for h-bonds/salt bridges in tables, and the BSA values should be rounded to the nearest 10.

As requested by the Reviewer, we have now removed Supplementary Table 2 that included information about H-bonds and salt bridges. BSA values have been corrected as follows:

- line 138: “Overall these motifs contribute 397 Å² (64%), 357 Å² (55%), 476 Å² (71%) and 227 Å² (28%) of surface area” by “Overall these motifs contribute 400 Å² (64%), 360 Å² (55%), 480 Å² (71%) and 230 Å² (28%) of surface area”.

-line 143: “ICOS-L buries 125 Å² of surface area” by “ICOS-L buries 130 Å² of surface area”

-Line 176: “buries approximately 424 Å² of BSA” by “buries approximately 420 Å² of BSA”

-Line 203: “805 to 622 Å², respectively” by “800 to 620 Å², respectively”

-Line 214: “902 Å² vs. 577 Å², respectively” by “900 Å² vs. 580 Å², respectively”

-Line 210: “BSA of 14 and 58 Å², respectively” by “BSA of 10 and 60 Å², respectively”

-Fig. 3

4) The statement relating to the CC' loop of ICOS being responsible for binding specificity to ICOS-L will need to be watered down or backed up formally with experimental data (e.g. mutational analyses)

To tone down the statement that the ICOS CC' loop is responsible for the binding specificity to ICOS-L, the sentence on line 252 has been re-worded to: “[...], ICOS utilizes a second set of residues within its CC' loop to contribute a considerable fraction of the contacts likely responsible for its binding specificity to ICOS-L”.

5) the ICOS-Fab structure was solved using the longer construct for ICOS. So, was this a disulfide-linked homodimer in the crystal structure, and does the monomeric/dimeric state of ICOS impact on Fab binding?

The ICOS/STIM003 complex was obtained using an ICOS construct encompassing residues 21-138. However, C136 and C137 were mutated to alanine residues. This clarifying detail is now stated in the Materials & Methods section on line 348: “Cys mutations were required to increase sample homogeneity and obtain well diffracting crystals.”

Regarding the effect of the monomeric/dimeric state of ICOS in STIM003 binding, we did not observe significant differences in binding affinities of the Fab to the monomeric or dimeric form of ICOS using BLI. The corresponding KD values measured are 0.9 and 2 nM, respectively.

Reviewer #2:

The ICOS/ICOS-L pathway plays important roles in T cell co-stimulation and T follicular helper (Tfh) cell generation. In addition, this pathway has been reported to be involved in cancers and autoimmune diseases. People in the field have been waiting for the structure of ICOS and/or ICOS-L, now Julien and his colleagues reported structural characterization of the ICOS/ICOS-L immune complex and ICOS-mAb complex. The structure revealed several new discoveries, including that a central FDPPPF motif and residues within the CC' loop of ICOS are responsible for the interaction with the ligand, N-linked glycosylation of ICOS 32 N110 is a gating element for ICOS-L binding, distinct receptor binding orientation compared to CTLA-4/B7-1 and PD-1/PD-L1. Finally, the authors showed that two therapeutic anti-ICOS mAbs bound ICOS similar to the receptor-ligand binding core interactions.

Overall, the manuscript is well-written and the structures are new and important. That being said, this Reviewer did not have a major concern. The below minor issues can be fixed to further enhance the quality of the manuscript:

We thank the Reviewer for the positive assessment of the manuscript and its impact on the field.

1) It is well accepted that ICOS proteins are expressed as homodimer on immune cell surface. However, the structure of ICOS in this manuscript is monomer, due to the fact that only amino acid 21-129 of ICOS were used to generate proteins. This weakness should be discussed.

We are aware of the limitation caused by truncating the ICOS construct at residue 129 and hence of our structural data to provide information about the oligomeric assembly of ICOS and ICOS-L. In fact, the crystallographic arrangement of ICOS protomers observed in our crystal structure is not compatible with the formation of an intramolecular disulfide bond and consequently we conclude that such assembly is likely an artefact of crystal packing (lines 174- 183). In addition, the following paragraph in the Discussion further acknowledges such limitation implying that additional studies are required to define the oligomeric state of ICOS:

“Our crystal structure of the ICOS/ICOS-L monomeric complex did not allow to define a biologically-relevant dimer interface for the ICOS ectodomain; indeed, a mutation removing the interchain disulfide bond in ICOS was required to obtain well diffracting crystals, resulting in the ICOS ectodomain being monomeric in solution and limited information could be derived from the crystal packing interfaces.”

2) The format of all references should be unified. Some references missed page numbers.

We thank the Reviewer for this careful reading of our manuscript. The electronic page number (e000146) has now been added to Reference 50:

Sullivan, B. A. *et al.* Inducible T-cell co-stimulator ligand (ICOSL) blockade leads to selective inhibition of anti-KLH IgG responses in subjects with systemic lupus erythematosus. *Lupus Sci. Med.* 3, e000146 (2016).

In addition, the electronic page number and volume number of Reference 68 has been added:

Morin, A. *et al.* Collaboration gets the most out of software. *Elife* 2, e01456 (2013).

Reviewer #3:

Rujas et al. is really just simply a beautiful manuscript: concise yet complete, rigorously performed and interpreted, interesting, important, and beautifully illustrated. All needed information is included, the scope is pleasingly comprehensive, and the impact is manifest. As-is, the manuscript is written to be widely accessible and will hold interest for a broad swath of the community.

We thank the Reviewer for these very encouraging comments.

1) The error on the "STIM003/ICOSdimer" interaction in Extended Table 1 should be reported as " 2 ± 3 " instead of " 2.4 ± 3.3 "

The values have been corrected in Supplementary Table 1.

REVIEWERS' COMMENTS

Reviewer #1 (Remarks to the Author):

I am satisfied with the revision.

Reviewer #2 (Remarks to the Author):

All my concerns have been addressed.

Xingxing Zang